# Characterization of KH-560-Modified Jute Fabric/Epoxy Laminated Composites: Surface Structure, and Thermal and Mechanical Properties

**DOI:** 10.3390/polym11050769

**Published:** 2019-05-01

**Authors:** Xue Wang, Lihai Wang, Wenwen Ji, Quanling Hao, Guanghui Zhang, Qingkai Meng

**Affiliations:** College of Engineering and Technology, Northeast Forestry University, Harbin 150040, China; wangxue6025@gmail.com (X.W.); 15504622683@163.com (W.J.); wendymouse2008@163.com (Q.H.); 18846184429@163.com (G.Z.); 18363996646@163.com (Q.M.)

**Keywords:** KH-560 modification, jute fabrics, interfacial compatibility, crystallinity, thermogravimetric

## Abstract

In this study, jute fabrics were used to reinforce epoxy resin to prepare laminated composites. KH-560 silane coupling agent modification was used to improve the interfacial compatibility between fibers and epoxy. The effects of different immersion times (0 min, 10 min, 30 min, 60 min, 90 min, and 120 min) on the jute fiber’s element content, crystal structure, and thermal stability, and the mechanical properties of laminated composites were studied. X-ray diffractometry (XRD) analysis showed that the KH-560 modification improved the crystallinity index (CI) and crystallite sizes (CS) of jute fibers. Scanning electron microscopy (SEM) analysis of the tensile fracture surfaces revealed a thick epoxy on the modified pulled fiber surfaces. Fourier transform infrared spectroscopy (FTIR) and energy dispersive spectrometer (EDS) analysis identified the presence of silicon and C–O–Si/Si–O–Si cross-linked structures on the surface of modified jute fibers. These cross-linked structures improved the thermal stability and mechanical properties of the laminated composites. When the immersion time was 60 min, the CI, CS, tensile strength, tensile modulus, flexural strength, and flexural modulus of the modified samples were 42.39%, 3.62 nm, 34.6 ± 1.1 MPa, 2.11 ± 0.12 GPa, 83.7 ± 1.8 MPa, and 4.08 ± 0.12 GPa, respectively, which were better than that of unmodified and other modified composites.

## 1. Introduction

Natural fiber reinforced laminated composites have broad application prospects in the fields of automobile manufacturing, building construction, and transportation due to their low cost, low environmental impact, and good mechanical properties [1,2]. Laminated composites were prepared using hot pressing, manual layering, or vacuum-assisted resin transfer molding. Natural fibers, such as ramie, hemp [3], sisal [4,5], flax [6], and jute [7,8], have been used in the reinforcement of laminated composites.

However, the natural fibers are cellulose-based, composed of cellulose, hemicellulose, lignin, pectin, wax, and ash. The hydrophilic cellulose fibers contain a large number of hydroxyl groups. The hydrogen bond interactions between the molecular chains are strong, which is not conducive to the combination with the hydrophobic matrix and, thus, degrade the mechanical properties of the composites [9,10,11,12]. Therefore, in order to improve the interfacial compatibility between hydrophilic natural fibers and a hydrophobic matrix, it is necessary to first perform a hydrophobic treatment on natural fibers. The pretreatment methods include heat treatment, alkali treatment, acylation treatment, and silane coupling agent treatment [13].

The silane coupling agents are polyfunctional organic compound, which act as bond bridges between natural fibers and the matrix to improve the mechanical properties of composites [9,14,15]. Seki [16] showed that the tensile and flexure properties of 1% silane-treated jute fabric-reinforced thermoset composites were better than that of untreated composites, and that the silane treatment increased the interlaminar shear strength of composites. Silane coupling agent treatment reduced the water absorption of composites [17]. The silane coupling agent modified the reaction between the natural fibers and the matrix, which reduced the gaps between fibers and the matrix of laminated composites. KH-550-treated composites showed higher thermal conductivity and better electrical properties when compared with untreated composites [18]. Silane coupling agents also had obvious advantages in the modification of bamboo fibers [19,20,21], where the mechanical properties of 0.7% vinyl functional silane coupling agent modified bamboo fiber-reinforced composites were superior than that of 0.3% maleic anhydride and 0.3% acrylic acid-treated composites [19]. Few scholars have studied the effect of processing time of silane coupling agent on the thermal stability and mechanical properties of jute fabric-reinforced laminate composites. However, in composite materials, jute fabrics with higher strength and toughness are better at transmitting than short plant fibers. It is worth noting that, if the silane coupling agent treats the fiber for too long, it will easily undergo the condensation reaction by itself. This will affect the grafting effect of the plant fiber surface and the thermal stability and mechanical properties of laminated composites. Therefore, the time and mechanism used for silane coupling agents to modify plant fibers require systematic research. 

In this work, the silane coupling agent, KH-560, was used to treat jute fabrics for 10, 30, 60, 90, and 120 min. Effects of the treated and untreated jute fabric-reinforced epoxy composites on the interfacial compatibility and thermal stability were analyzed with the aid of scanning electron microscope (SEM), X-ray diffractometry (XRD), Fourier transform infrared spectroscopy (FTIR) and thermogravimetric analysis (TG/DTG). Flexural, tensile strength, and modulus of laminated composites were tested to assess the effects of immersion times on their mechanical properties.

## 2. Materials and Methods 

### 2.1. Materials

The silane coupling agent KH-560 (≥98%) and acetic acid were supplied by Harbin Sailebo Technology Ltd (Harbin, China). The chemical formula of KH-560 is shown in Figure 1. Jute fabrics were obtained from Guangzhou Yin Fan Textile Ltd (Guangzhou, China). Matrix materials (epoxy E-44, polyamide resin 650, and thinner 692) were supplied by Zhenjiang Danbao Resin Ltd (Zhenjiang, China). The polyamide resin 650 was the vegetable oil and ethylene amine dimer acted as the curing agent, which was mixed with epoxy. Thinner 692 was 1-benzyloxy-2,3-epoxypropane (C_10_H_12_O_2_), which was characterized by low viscosity, easy dilution, and low volatility. The mass ratio of E-44:polyamide resin 650:thinner 692 was 10:6:1.

### 2.2. Fabrication of Laminated Composites

#### 2.2.1. Surfaces Treatment for Jute Fabrics

A 2% (in weight) KH-560 silane coupling agent solution was prepared. The pH of the KH-560 solution was adjusted to 4 by adding an appropriate amount of acetic acid to promote KH-560 hydrolysis. Jute fabrics (size: 27 cm × 15 cm) were immersed in the 2% silane coupling agent KH-560 for 10, 30, 60, 90, and 120 min at room temperature. Then the jute fabrics were dried in an oven at 80 °C for 5 h. 

The structural formula of KH-560 contains an epoxy group that acts as an organofunctional silane, which can improve the interfacial compatibility between jute fabric and epoxy resin. The mechanism of surface modification of the silane coupling agent is shown in Figure 1. In the first step, the methoxy groups (Si–O–CH_3_) in KH-560 hydrolyze to the hydroxyl groups (Si–OH) in the aqueous solution. In the second step, the hydroxyl groups formed by the first step react with the hydroxyl groups on the surface of the jute fiber and they connect via hydrogen bonds. In the third step, Si–O–Si or C–O–Si groups are formed between the silane coupling agents (Si–OH) and the hydroxyl groups (jute–OH) on the fiber surface in the dehydration condensation process.

#### 2.2.2. Laminated Composites Processing

As shown in Figure 1, the silane coupling agents are equivalent to “bond bridges” between jute fabrics and epoxy resins that improve the mechanical properties of laminated composites. Jute fabrics-reinforced laminated composites were fabricated by manual layering. First, epoxy resin E-44, polyamide resin 650 (curing agent), and thinner 692 were uniformly mixed at a ratio of 10:6:1 (in weight) and applied to the jute fabrics. Silicone cushions were the separator material to avoid composites bonding with other objects (trays, tables, and ovens, etc.). Two layers of modified jute fabrics and epoxy resins were laid between the silicone cushions (Figure 2). The fiber content was 20% (in weight). Then, the laminated composites were cured in a drying oven for 1.5 h at 120 °C. A steel plate, weighing about 1 kg, was placed on top of the laminated composite to apply pressure. Finally, the cured laminated composites were taken out and the silicone cushions were removed after cooling for 10 min. The KH560 modified and unmodified jute fabric-reinforced laminated composites were prepared using the above methods.

### 2.3. Characterization

#### 2.3.1. Fourier Transform Infrared (FTIR) Spectroscopy

In order to evaluate the effects of KH-560 modification on the surface functional groups of jute fabric, FTIR analysis was carried out using Thermo Nicolet FTIR spectrometer (Thermo Fisher Scientific, Waltham, Massachusetts, United States) from 4000 cm^−1^ to 500 cm^−1^. The modified and unmodified jute fabrics were cut into pieces and mixed with KBr and then compressed.

#### 2.3.2. X-ray Diffractometry (XRD)

The crystallinity index (CI) and crystallite sizes (CS) of modified and unmodified jute fabrics were characterized using X-ray diffractometry (X’Pert Power, PANalytical, Almelo, Netherlands). The scanning range was 10°–90°. And the CI and CS were calculated using Equations (1) and (2), respectively [22]:(1)$$\mathrm{CI}=\left(1-\frac{I_{am}}{I_{002}}\right)\times 100\%,$$
(2)$${\mathrm{CS}}_{002}=\frac{0.89\lambda }{\beta_{002}\cos\theta },$$where *I*_002_ and *I_am_* are the intensity of the crystalline phase peak around 23° and the amorphous phase peak around 16°, respectively. *λ* (0.154 nm) is the wavelength of the X-ray, *β*_002_ is the full width half maxima, and *θ* is the angle of diffraction.

#### 2.3.3. Thermogravimetric Analysis (TG/DTG)

In order to study the thermal degradation of KH-560 modified jute fabrics reinforced laminated composites. Thermogravimetric analyses (TG/DTG) were conducted using a thermal gravimetric analyzer (STA449 F3, NETZSCH Scientific, Selb, Germany). The unmodified and KH-560-modified jute fabric-reinforced composites were heated from 23 °C to 800 °C at a rate of 10 °C/min.

#### 2.3.4. Energy Dispersive Spectrometer (EDS)

The elements on the surface of the jute fiber before and after modification with the KH-560 silane coupling agent were tested by energy-dispersive spectrometer (EDS, JEC-3000FC, Inca X-Max, Oxford Instruments, Oxford, Britain). The contents of the C, O, and Si elements on the surface of the modified and unmodified jute fibers were tested.

#### 2.3.5. Scanning Electron Microscope (SEM)

The morphological changes of the fibers’ surface and tensile fracture surface of the laminated composite before and after KH-560 treatment were observed using SEM JSM-7500F (JEOL Ltd., Akishima, Tokyo, Japan). All samples were coated with a thin layer of gold prior to SEM observation.

### 2.4. Mechanical Properties

#### 2.4.1. Tensile Property

The tensile properties of the laminated composites were tested by GB/T 1447-2005 with a test speed of 5 mm/min [23]. Ten bars were tested of each sample and the average values were reported. The sample size was 3 mm × 25 mm × 250 mm, the size of the end wood reinforcing sheet was 2 mm × 25 mm × 50 mm, and the gauge length was 100 mm.

#### 2.4.2. Flexural Property

The three-points bending properties of the KH-560 modified and unmodified jute fabrics reinforced laminated composites were tested according to GB/T 1449-2005 at a test speed of 2 mm/min [24]. Ten bars of each sample were tested and the average values were reported. The sample size was 3 mm × 25 mm × 125 mm and the span was 80 mm.

## 3. Results and Discussion

### 3.1. FTIR Analysis of KH-560 Modified and Unmodified Jute Fabrics

FTIR spectra of unmodified and KH-560-modified jute fabrics, following different time durations, are shown in Figure 3 from wavenumber 4000 cm^−1^ to 500 cm^−1^. The peak positions and the chemical composition assignments of unmodified and modified jute fabrics are tabulated in Table 1. The absorption bands at 3329 cm^−1^ and 2917 cm^−1^ for unmodified jute fabrics were associated with -OH and C–H stretching vibration in the cellulose, respectively [22,25]. But, the relative peaks occurred at 3336 cm^−1^ and 2938 cm^−1^, which became weaker after KH-560 treatment. The reason for this is that the -OH groups of cellulose were modified by KH-560 to form hydrogen bonds, which reduced the vibration of both C–H bonds and -OH groups because C–H bonds share a carbon with the modified -OH groups in the β(1–4)-linked D-glucose units of cellulose. The -OH groups of cellulose were preferably modified by KH-560 because the content of cellulose, hemicellulose, and lignin in jute fiber is about 54%, 26% and 11%, respectively, and the number of hydroxyl groups on cellulose is greater than on hemicellulose and lignin. The peaks at 1735 cm^−1^, 1653 cm^−1^, 1421 cm^−1^, and 1235 cm^−1^ were attributed to the C=O stretching of carboxylic acid in hemicellulose, the C=C stretching of oils on the jute surface, CH_2_ stretching of cellulose, and the C–O stretching of lignin, respectively [26,27,28,29]. The KH-560 surface modification reduced the intensity of vibration peaks at 1653 cm^−1^ and 1235 cm^−1^ because the oils on the surface of the jute fiber were removed by acidic solution soaking, while part of the lignin were covered by the coupling agent. The vibration peak at 1023 cm^−1^ was due to the C–OH stretching vibration in lignin [30,31], which was reduced after the KH-560 modification at 10, 30, and 60 min. The Si–OH stretching vibration was more pronounced at 986 cm^−1^ (S2, S3, S4). It is worth noting that the new vibration peaks at 986 cm^−1^ and 1105 cm^−1^ were formed by Si–OH stretching vibration and C–O–Si/Si–O–Si stretching vibration [32]. The vibration intensity of the jute fabric modified by KH-560 for 60 min was the highest at both 986 cm^−1^ and 1105 cm^−1^. The FTIR analysis indicated that KH-560 successfully modified jute fabrics and formed C–O–Si functional groups on the fiber surface by dehydration condensation between Si–OH and hydroxyl groups. 

### 3.2. XRD Analysis of KH-560 Modified and Unmodified Jute FABRICS

X-ray diffractometry (XRD) was used to analyze the effect of KH-560 silane coupling agent modification on the crystallinity of jute fabric. As shown in Figure 4a, the diffraction peaks at 15.6° and 23.3° are the 101 and 002 crystal faces of the cellulose structure, respectively [33]. The fine crystal form of cellulose consists of the crystalline region and the amorphous region, which form the different diffraction peaks at 15.6° and 23.3°. KH-560 modified the fiber surface with the -OH groups of cellulose, which improved the crystallinity index (CI) and crystallite size (CS) without damaging the fine crystal structure of jute fiber. Figure 4b shows that the crystallinity index (CI) and crystallite size (CS) of the modified jute fibers (S2–S6) were improved by different degrees than that of unmodified jute fiber (S1). Notably, the CI and CS of the 60 min KH-560-modified fiber increased from 37.07% and 3.58 nm to 42.39% and 3.62 nm, respectively. The optimal immersion time of the jute fabric was 60 min in the 2% KH-560 solution with a pH of 4 at room temperature (24 °C). The reason for this was that if the immersion times were too long (90 min and 120 min), the hydrolyzed KH-560 was liable to agglomerate by itself. Too short of an immersion (10 min and 30 min) also affected the number of Si–OH groups grafted onto the jute fabric. The EDS analysis (Figure 5 and Table 2) also showed that the Si element had the highest content (weight: 1.53%, molar: 0.72%) on the jute fiber surface when the immersion time was 60 min. The XRD analysis showed that KH-560 modification improved the CI and CS of jute fabrics, and the optimal immersion time was 60 min.

### 3.3. SEM and EDS Analysis of KH-560-Modified and Unmodified Jute Fabrics

Figure 5 shows the surface morphology and energy spectrum of the jute fiber before and after modification with the KH-560 silane coupling agent. The SEM images of Figure 5a–f show that there were no significant changes between the modified jute fibers and unmodified jute fiber. This indicates that the silane coupling agent modification did not damage the surface of the jute fiber. It can be seen from the EDS that the unmodified jute fiber (S1-0 min) contained a very small amount of Si, and that the surfaces of jute fibers modified with KH-560 (S2–S6) contain different contents of Si. This indicates that KH-560 successfully grafted to the surface of jute fibers. Table 2 shows the weight percentages and molar percentages of C, O and Si contained on the surface of modified and unmodified jute fibers. The results show that carbon and oxygen were the main peaks in unmodified and modified jute fibers. This is because they are the main elements of cellulose, hemicellulose, and lignin in jute fibers [22,31]. A small amount of Si element was present on the surface of the unmodified jute fiber, and its weight percentage and molar percentage were 0.15% and 0.07%, respectively. However, the contents of Si on the surface of modified fibers increased after KH-560 modification. When the immersion time was 60 min, the weight percentage and molar percentage of Si increased to 1.53% and 0.72% respectively, which were 10-times greater that of unmodified jute fiber.

### 3.4. TG and DTG Analysis of Laminated Composites

Thermal stability is an important property of the fiber composites [34]. Figure 6 and Table 3 show the degree of weight loss for the laminated composites before and after KH-560 treatment. The TG and DTG curves showed that the weight and weight loss rate of the laminated composites exhibited four stages of change with increasing temperature. The temperature ranges of the untreated laminate composite S1 were 22–100 °C, 100–230 °C, 230–470 °C, and 470–790 °C, respectively. After the KH-560 treatment, the second temperature stage of the laminated composite increased by ≥35 °C, and the third stage decreased by ≥35 °C. The weight loss in the first, second, third, and fourth stages was due to the moisture absorbed by the jute fiber (0.8%–1.5%); the degradation of lignin in the jute fiber (2.8%–3.4%); the decomposition of cellulose, hemicellulose, and epoxy resin (80.7%–88.2%); and the carbonization of the laminated composite (4.2%–7.1%), respectively. The weight loss in the third stage was the greatest, which is the main stage of thermal degradation of laminated composites. Figure 1 and FTIR analysis illustrated that the C–O–Si functional groups were formed between the KH-560-modified jute fibers and the epoxy resin. The connection between the unmodified jute fiber and the epoxy resin was the mechanical bite force rather than chemical bonds. The interface between the epoxy resin and the KH-560-modified jute fiber surface was a network of cross-linked molecular layers (C–O–Si/Si–O–Si), both single and multi-layer. Therefore, the cross-linked structures absorbed more energy than the mechanical bite force during the degradation process, requiring higher temperatures to destroy the C–O–Si groups on the cellulose and hemicellulose surfaces. 

All of the KH-560-modified laminated composites had greater residuals than that of the unmodified composites. The residual weights at S3 and S4 were 10.1% and 9.4%, respectively, which was about ten-times that at S1. This was because KH-560 forms a “protective net” on the surface of the jute fibers, so that fibers were protected at high temperature [35]. Therefore, the modified laminated composites had higher carbon contents than that of unmodified laminated composite. The above results showed that the KH-560 silane coupling agent improved the thermal stability of laminated composite and that the composites with an immersion times of 30 min and 60 min had better thermal stability.

### 3.5. Mechanical Properties and SEM Analysis of Laminated Composites

Jute fiber is a hydrophilic cellulose fiber with a large amount of hydroxyl groups. The epoxy resin is a thermosetting polymer material with hydrophobicity. Unmodified jute fibers mechanically interlock with the epoxy resin. The cross-linked structures of C–O–Si and Si–O–Si were formed between the KH-560-modified jute fibers and the epoxy resins. The cross-linked structures can absorb more energy than the mechanical bite, when forces try to damage the laminated composites. Table 4 demonstrates that KH-560-modified laminated composites had greater strengths and modulus than that of the unmodified laminate composites. The flexural strength and modulus were higher than the tensile strength and modulus. The reason for this is that the strong cross-linking over Si–O–Si might make the molecular layer between the jute fiber surface and the epoxy more inflexible. When the immersion time was 60 min, the tensile strength, tensile modulus, flexural strength, and flexural modulus of the composite increased by 42%, 39%, 49%, and 51%, respectively. These values were higher than that of other modified and unmodified composites and wood-plastic composites (Table 4) [36,37,38,39]. Bengtsson and Oksman [39] added silane coupling agents to softwood powder and high-density polyethylene to improve the interface compatibility. The flexural strength and modulus of modified wood-plastic composites were 49.9 ± 0.2 MPa and 2.6 ± 0.1 GPa, respectively. The strength of composites is affected by their density, moisture content, fiber content, and strength of the matrix, as well as the length of fibers. Wood-plastic composites are usually reinforced with wood powder or short natural fibers. In this study, jute fabrics were used as the reinforcement and had better strength and toughness. They acted as a good force transfer in the composites.

Figure 7S1–S6 is the SEM analysis of tensile fracture surfaces of the laminated composites treated with the silane coupling agent at different immersion times. Figure 7a–c are partial enlarged electron microscope images of S1, S2, and S3, respectively. The unmodified composite, S1, showed that the tensile failure mode was the fiber pull-out, and there were many holes in the fracture surface, and the pull-out fibers (Figure 7a) of the unmodified composites were not coated with a layer of epoxy. SEM for S2–S6 of the KH-560-modified composites showed that the jute fibers were better dispersed in the epoxy resin, which increased the contact areas between jute fibers and the epoxy resins. It was obvious that the pull-out fibers of modified composites were covered with a thick layer of epoxy resin (Figure 7b,c). There were fine fiber filaments on the pull-out fibers of Figure 7a compared to Figure 7b,c. The reason for this was that the KH-560 modification enhanced the interface between epoxy and fiber. Therefore, the fiber was wrapped with epoxy resin when it was pulled out. This was due to the cross-linked structures (C–O–Si and Si–O–Si) between jute fabrics and epoxy resins after KH-560 modification, so that the force could form the fiber-resin-fiber transmission. The above results indicated that KH-560 modification improved the interfacial compatibility and mechanical properties of the laminated composites.

## 4. Conclusions

The above studies showed that KH-560 was successfully grafted onto the surface of jute fibers. The silicon content of the modified jute fiber was detected in EDS analysis to be about 10-times greater than that of the unmodified jute fiber. C–O–Si/Si–O–Si (1105 cm^−1^) and Si–OH (986 cm^−1^) functional groups of the modified jute fibers were detected in the FTIR analysis. These characteristic functional groups formed cross-linked structures on the surface of jute fibers, which not only improved the interface compatibility between jute fiber and epoxy resin, but also increased the crystallinity and thermal stability of the jute fibers. Laminated composites with an immersion time of 60 min had the best mechanical properties: the tensile strength, tensile modulus, bending strength, and flexural modulus were enhanced by 42%, 39%, 49% and 51%, respectively, which was better than many wood-plastic composites. SEM analysis of the tensile fracture surface showed that the failure mode of the laminated composites was fiber pulling out. The surface of the unmodified pulled fibers was clean and there were fine fiber filaments on the pulled-out fibers, while the surface of the modified pulled fibers was covered with a layer of epoxy resin. The wrapped fibers formed a protective barrier on the surface during the heating process, which was beneficial to improve the thermal stability of the laminated composites. The maximum degradation stage of the laminated composites was during the third stage (>80%), and the temperature with the maximum weight loss rate appeared in this stage. Laminated composites with an immersion times of 30 min and 60 min had higher residual weight and better thermal stability. The KH-560-modified jute fabric-reinforced laminated composites had good mechanical properties and thermal stability, which could be used in the preparation of sandwich structures in wood structures. This method could also be used for reference in the preparation of automotive interior green composites.

## Figures and Tables

**Figure 1 polymers-11-00769-f001:**
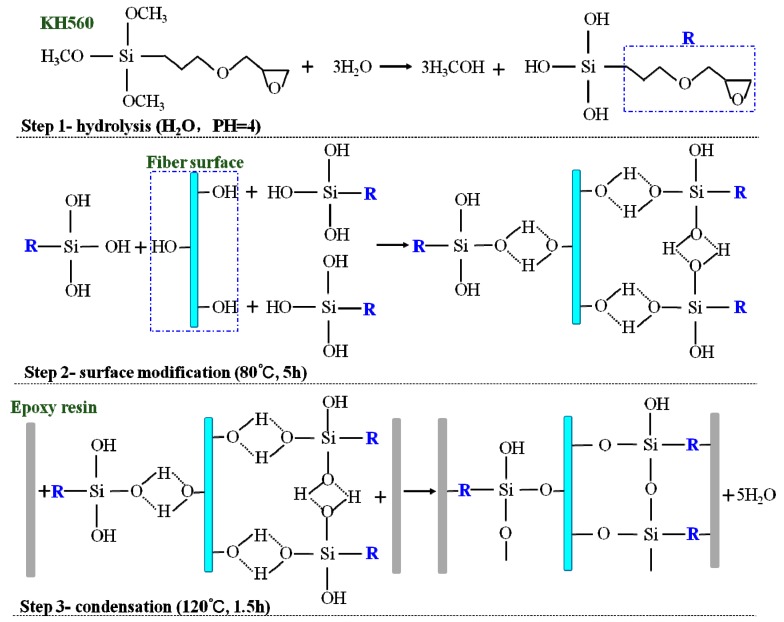
The schematic diagram of the KH-560 silane coupling agent modifying the surface of jute fabric.

**Figure 2 polymers-11-00769-f002:**
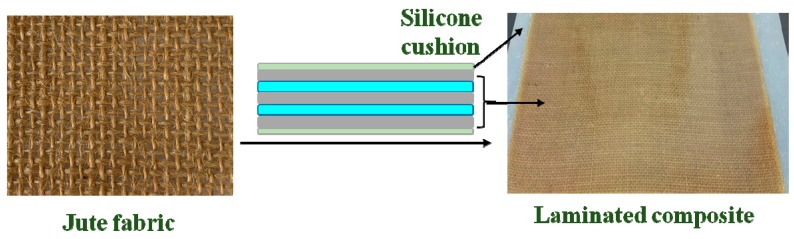
The KH-560-modified jute fabrics reinforced laminated composites.

**Figure 3 polymers-11-00769-f003:**
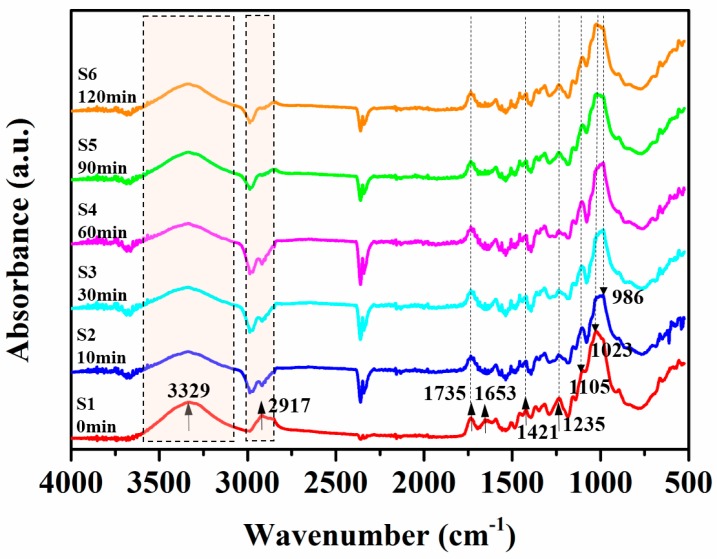
Fourier transform infrared (FTIR) spectra of unmodified and KH-560-modified jute fabrics.

**Figure 4 polymers-11-00769-f004:**
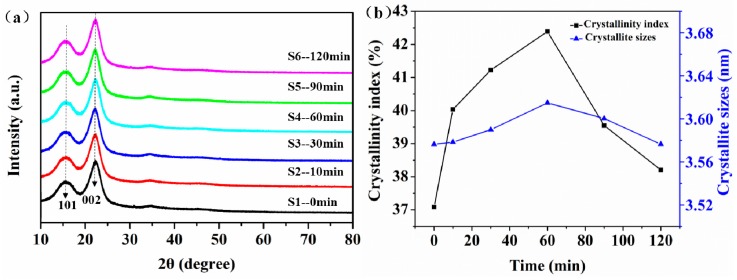
(**a**) X-ray Diffractometry (XRD) analysis and (**b**) crystallinity index (CI) and crystallite size (CS) of unmodified and modified jute fabrics.

**Figure 5 polymers-11-00769-f005:**
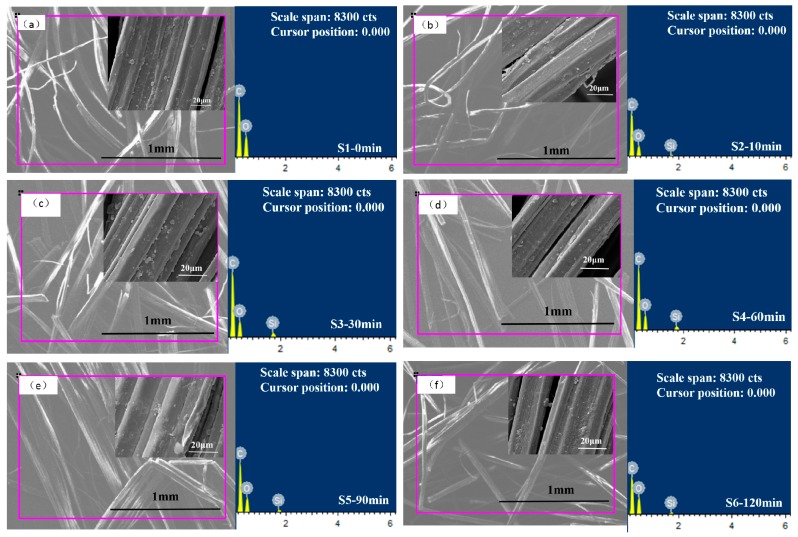
SEM and energy dispersive spectrometer (EDS) analysis of (**a**) unmodified jute fiber and (**b**–**f**) KH-560-modified jute fibers.

**Figure 6 polymers-11-00769-f006:**
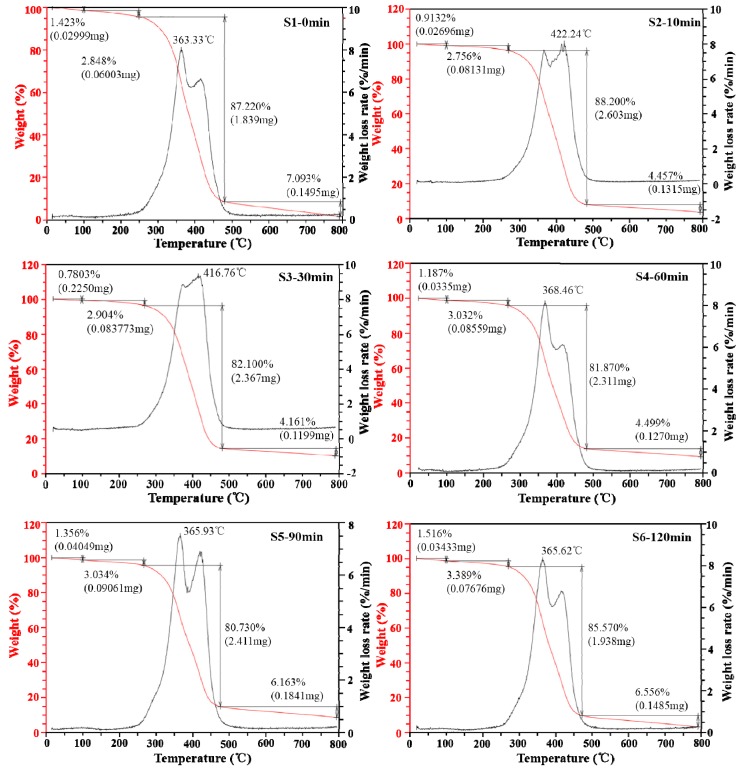
Thermogravimetric (TG) and DTG curves of unmodified and KH-560-modified jute fabrics and epoxy composites. (S1–S6 are samples with immersion times of 0, 10, 30, 90, and 120 min, respectively).

**Figure 7 polymers-11-00769-f007:**
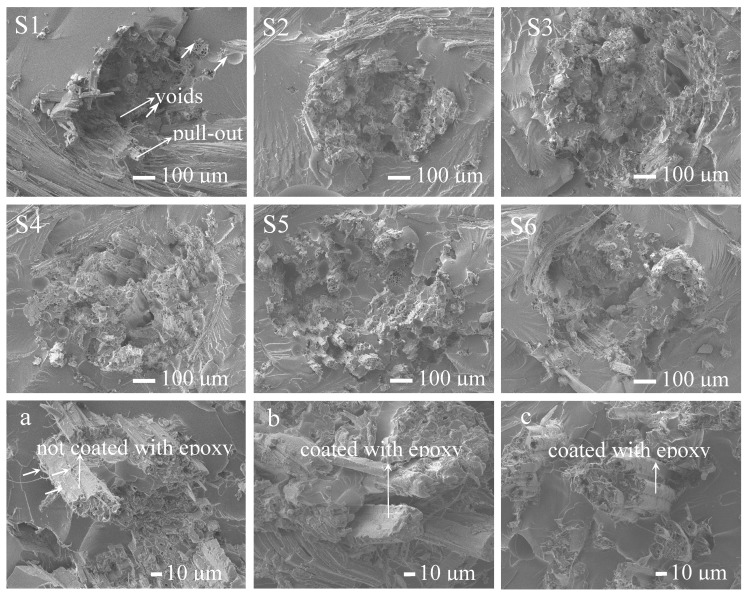
SEM analysis of tensile fracture surfaces: (**S1**–**S6**) are the tensile fracture surfaces of the laminated composite materials with immersion times of 0, 10, 30, 60, 90, and 120 min, respectively. (**a**–**c**) are the partial enlarged electron microscope images of S1, S2 and S3 respectively.

**Table 1 polymers-11-00769-t001:** Peak positions and assignments in FTIR spectra of unmodified and KH-560-modified jute fabrics following different KH-560 modification time durations.

Assignments	Peak Positions Wavenumber (cm^−1^)						Reference
	S1	S2	S3	S4	S5	S6	
-OH stretching of cellulose	3329	3336	3336	3336	3336	3336	[22]
C–H stretching of cellulose	2917	2938	2938	2938	2938	2938	[25]
C=O stretching of hemicellulose	1735	1735	1735	1735	1735	1735	[26]
C=C stretching of oils	1653	--	--	--	--	--	[27]
CH_2_ stretching of cellulose	1421	1420	1420	1420	1420	1420	[28]
C–O stretching of lignin	1235	1237	1237	1237	1237	1237	[29]
C–O–Si and/or Si–O–Si stretching	--	1105	1105	1105	1105	1105	[30,31]
C–OH and/or Si–OH stretching	1023	986	986	986	1023	1023	[31,32]

**Table 2 polymers-11-00769-t002:** The elements and their contents for unmodified and KH-560-modified jute fibers.

Sample Numbers	C (%)		O (%)		Si (%)	
	Weight	Molar	Weight	Molar	Weight	Molar
S1-0min	57.51	64.36	42.33	35.57	0.15	0.07
S2-10min	63.48	69.98	35.96	29.76	0.56	0.26
S3-30min	57.07	64.19	41.72	35.23	1.20	0.58
S4-60min	64.29	70.95	34.18	28.32	1.53	0.72
S5-90min	55.39	62.52	43.72	37.05	0.89	0.43
S6-120min	62.37	69.04	36.75	30.54	0.89	0.42

**Table 3 polymers-11-00769-t003:** TG and DTG data from thermogravimetric analysis of unmodified and KH-560-modified jute fabrics and epoxy composites.

Sample Numbers	First Stage T_range_ 22–100 ℃	Second Stage		Third Stage		Fourth Stage T_range_ 470–790 ℃	Residual Weight (%)
	W_loss_ (%)	T_range_ (℃)	W_loss_ (%)	T_range_ (℃)	W_loss_ (%)	W_loss_ (%)	
S1-0min	1.423	100−230	2.848	230−470	87.220	7.093	1.453
S2-10min	0.913	100−265	2.756	265−480	88.200	4.457	3.673
S3-30min	0.780	100−265	2.904	265−480	82.100	4.161	10.120
S4-60min	1.187	100−265	3.032	265−480	81.870	4.499	9.419
S5-90min	1.356	100−265	3.034	265−475	80.730	6.163	8.721
S6-120min	1.516	100−270	3.389	270−470	85.570	6.556	2.965

T_range_: the temperature range; W_loss_: the weight loss.

**Table 4 polymers-11-00769-t004:** Mechanical properties of unmodified and KH-560-modified jute fabric and epoxy composites.

Sample Numbers	Densities of Composites (g﹒cm^−3^)	Tensile Strength (MPa)	Tensile Modulus (GPa)	Fracture Strain (%)	Flexural Strength (MPa)	Flexural Modulus (GPa)
S1-0min	1.09 ± 0.03	24.3 ± 1.1	1.52 ± 0.05	1.56 ± 0.18	56.0 ± 2.5	2.71 ± 0.17
S2-10min	1.07 ± 0.03	27.0 ± 1.2	1.71 ± 0.02	1.58 ± 0.15	74.8 ± 2.2	3.64 ± 0.18
S3-30min	1.08 ± 0.02	33.8 ± 1.1	2.09 ± 0.05	1.62 ± 0.17	76.9 ± 2.4	3.75 ± 0.14
S4-60min	1.12 ± 0.01	34.6 ± 1.1	2.11 ± 0.12	1.64 ± 0.19	83.7 ± 1.8	4.08 ± 0.12
S5-90min	1.15 ± 0.02	30.3 ± 1.1	1.86 ± 0.10	1.63 ± 0.17	80.3 ± 1.2	3.91 ± 0.21
S6-120min	1.14 ± 0.01	29.6 ± 1.2	1.9 ± 0.09	1.56 ± 0.16	75.9 ± 2.3	3.51 ± 0.26
[34]	1.511	10.3	--	--	40.5	3.56
[35]	0.997 ± 0.02	19.3 ± 0.8	2.35 ± 0.06	1.50 ± 0.2	24.9 ± 1.3	1.81 ± 0.04
[36]	--	12.5 ± 0.5	3.41 ± 0.10	--	16.0 ± 1.0	2.94 ± 0.16
[37]	--	--	--	--	49.9 ± 0.2	2.60 ± 0.10

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
