# Peer review of "Characterization of KH-560-Modified Jute Fabric/Epoxy Laminated Composites: Surface Structure, and Thermal and Mechanical Properties"

_polymers, 2019, doi:10.3390/polym11050769_

Round 1

Reviewer 1 Report

Manuscript ID: polymers-488274

Authors: Xue Wang; Lihai Wang; Wenwen Ji; Quanling Hao; Guanghui Zhang; Qingkai Meng

Title: Characterization of KH-560 modified jute fabrics/epoxy laminated composites: surface structure, thermal and mechanical properties

The paper entitled “Characterization of KH-560 modified jute fabrics/epoxy laminated composites: surface structure, thermal and mechanical properties” by Lihai Wang and his coworkers reported an interesting experimental and modelling discussion in the modification and enhanced thermal and mechanical properties of jute fabrics/epoxy laminated composites by using silane couping agent KH-560. The finding is significant in manipulating the composites. It demonstrates the use of simple modification process using KH-560 to achieve high performance polymer composites.

This manuscript should be revised according to the following comments:

1. In Figure 1, the chemical structures are incorrect and unclear. Please revise it.

2. In Figure 5, the diagrams should be illustrated in details. Please modify it.

3. The applications of such a work ought to be better mentioned.

4.     The authors refer to the crystallinity index (CI) and crystallite sizes (CS) of modified and unmodified jute fabrics that can be calculated by the X-ray diffractometry. Citation should be provided.

Overall, the paper shows interesting findings, but minor revision should be made before publication.

Author Response

Dear Reviewer,    

Thank you very much for your constructive comments and suggestions to our manuscript entitled “Characterization of KH-560 modified jute fabrics/epoxy laminated composites: surface structure, thermal and mechanical properties” (488274). We are sure that your comments and suggestions are very important to improve the quality of our work. We have considered all your suggestions carefully. The following is the detailed responses to comments. 

Thank you and best regards. 

Yours sincerely, 

Prof. Lihai Wang College of Engineering and Technology, Northeast Forestry University. Corresponding author: Lihai Wang    Tel: +86 451 82190671 

E-mail: wanglihai@nefu.edu.cn (L.W.); wangxue6025@gmail.com (X.W.).

Reviewer 2 Report

Questions and comments see attached file.

Author Response

Dear Reviewer:

Thank you very much for your comments concerning our manuscript entitled “Characterization of KH-560 modified jute fabrics/epoxy laminated composites: surface structure, thermal and mechanical properties” (488274). Those comments are all valuable and very helpful for revising and improving our paper, as well as the important guiding significance to our researches. We have studied comments carefully and have made correction, which we hope meet with approval. Revised portions are marked in red in the paper. The main corrections and the responds are as follows.

Thank you and best regards.

Yours sincerely,

Prof. Lihai Wang

College of Engineering and Technology, Northeast Forestry University.

Corresponding author: Lihai Wang    Tel: +86 451 82190671

E-mail: wanglihai@nefu.edu.cn (L.W.); wangxue6025@gmail.com (X.W.).
